# PROTOCOL: Partial Optimal Transport-enhanced Contrastive Learning for Imbalanced Multi-view Clustering

Xuqian Xue [1]   Yiming Lei [2]   Qi Cai [3]   Hongming Shan [4]   Junping Zhang [1]

## Abstract

While contrastive multi-view clustering has achieved remarkable success, it implicitly assumes balanced class distribution. However, real-world multi-view data primarily exhibits class imbalance distribution. Consequently, existing methods suffer performance degradation due to their inability to perceive and model such imbalance. To address this challenge, we present the first systematic study of imbalanced multi-view clustering, focusing on two fundamental problems: *i. perceiving class imbalance distribution*, and *ii. mitigating representation degradation of minority samples*. We propose PROTOCOL, a novel PaRtial Optimal TranspOrt-enhanced COntrastive Learning framework for imbalanced multi-view clustering. First, for class imbalance perception, we map multi-view features into a consensus space and reformulate the imbalanced clustering as a partial optimal transport (POT) problem, augmented with *progressive mass constraints* and *weighted KL divergence* for class distributions. Second, we develop a POT-enhanced class-rebalanced contrastive learning at both feature and class levels, incorporating *logit adjustment* and *class-sensitive learning* to enhance minority sample representations. Extensive experiments demonstrate that PROTOCOL significantly improves clustering performance on imbalanced multi-view data, filling a critical research gap in this field.

[1]Shanghai Key Laboratory of Intelligent Information Processing, College of Computer Science and Artificial Intelligence, Fudan University, China. [2]College of Computer Science and Technology, Qingdao University, China. [3]Shanghai Key Laboratory of Navigation and Location-based Services, School of Electronic Information and Electrical Engineering, Shanghai Jiao Tong University, China. [4]Institute of Science and Technology for Brain-Inspired Intelligence and Key Laboratory of Computational Neuroscience and Brain-Inspired Intelligence, Fudan University, China. Correspondence to: Junping Zhang <jpzhang@fudan.edu.cn>, Yiming Lei <leiyiming@qdu.edu.cn>.

*Proceedings of the $42^{nd}$ International Conference on Machine Learning*, Vancouver, Canada. PMLR 267, 2025. Copyright 2025 by the author(s).

## 1. Introduction

Multi-view clustering (MVC) (Fang et al., 2023; Peng et al., 2019) aims to group samples by integrating complementary information across different views, inspired by humans' ability to perceive their environment through multi-sensory integration (Xu et al., 2013; Chidambaram et al., 2023). With the advancement of deep learning, Deep MVC (DMVC) (Chen et al., 2022; Liu et al., 2023) has emerged as the dominant paradigm for its superior performance.

Contrastive multi-view clustering, which combines contrastive learning with K-means, has become the leading approach in DMVC (Lin et al., 2023b; Li et al., 2022). By treating multi-view features as positive pairs, this approach has achieved significant advances in semantic consistency (Xu et al., 2022; Tian et al., 2020; Zhang et al., 2024c), feature discriminability (Lin et al., 2023b; Yan et al., 2023; Trosten et al., 2021), and view difference robustness (Xu et al., 2023; Trosten et al., 2023). However, these methods implicitly assume balanced class distribution, which may not hold in real-world multi-view data where class imbalance is prevalent. For example, in wildlife monitoring systems (Pan et al., 2024), multiple cameras capture rich visual information about animals, yet rare species (*e.g.*, Siberian tigers) are significantly underrepresented compared to common ones (*e.g.*, wild rabbits). Consequently, existing methods suffer significant performance degradation due to their inability to perceive and model such imbalance (Zhang et al., 2023b; Han, 2023).

These observations lead us to think the following question:

> *How to effectively perceive and model class imbalance distribution under unsupervised multi-view settings?*

To tackle this challenge, we decompose it into two fundamental scientific questions. First, we address *Q1: how to perceive class imbalance distribution.* Our method draws inspiration from two domains: supervised imbalanced classification (Cui et al., 2024; Suh & Seo, 2023; Wang et al., 2023) that leverages ground-truth labels, and self-labeling methods (Asano et al., 2020; Tai et al., 2021; Chapel et al.,

2020) that automatically assign labels during representation learning. This inspires us to optimize self-labeling to generate pseudo-labels that align with imbalance class distribution. We propose an imbalance-aware multi-view framework that combines POT with unbalanced optimal transport (UOT) to generate POT labels. This framework enables dynamic transportation from sample distribution to the underlying imbalanced class distribution while effectively integrating information across different views.

Building upon our solution to *Q1*, we address a second critical challenge *Q2: how to mitigate representation degradation of minority samples*. When using imbalanced POT labels, contrastive learning tends to bias towards dominant classes, leading to degraded representations of minority samples (Suh & Seo, 2023). While supervised methods like resampling (Gao et al., 2023), class-sensitive learning (Han, 2023), and logit adjustment (Cui et al., 2021) can address this issue using ground-truth labels, our unsupervised setting requires a different approach. To this end, we leverage POT labels as weak supervision signals and develop a two-level rebalancing strategy: logit adjustment at the feature level and class-sensitive learning at the class level, effectively enhancing the representations of minority samples.

While extensive research explored DMVC, the challenge of class imbalance has been overlooked, limiting its practical applications. To bridge this gap, we propose an imbalanced contrastive MVC method with three key stages. *First*, we learn view-specific representations through reconstruction loss. *Second*, we achieve semantic consistency through multi-view contrastive learning. *Finally*, we address class imbalance through two complementary components: *imbalance perception* and *imbalance modeling*. For imbalance perception, we reformulate the clustering problem by combining unbalanced OT with partial transport constraints, generating POT labels through progressive assignment based on transport costs and dynamic mass adjustment. For imbalance modeling, we leverage these POT labels to adjust the model's output distribution, effectively mitigating the representation degradation of minority samples.

**Our main contributions** are summarized as follows:

- We propose PROTOCOL, a novel imbalanced contrastive multi-view clustering method with three-stage collaborative optimization: view-specific feature learning, consensus representation learning, and imbalanced distribution modeling. Extensive experiments demonstrate its superior performance over existing methods.

- We propose a novel multi-view POT label assignment method that integrates partial optimal transport with unbalanced optimal transport to effectively perceive underlying class imbalance distributions under unsupervised multi-view setting.

- We propose a POT-enhanced class rebalanced contrastive learning method for imbalanced multi-view clustering, which mitigates representation degradation of minority samples through feature-level logit adjustment and class-level class-sensitive learning.

## 2. Related Work

### 2.1. Deep multi-view clustering

Deep multi-view clustering primarily encompasses two directions based on data completeness: Deep complete Multi-View Clustering (DMVC) and Deep Incomplete Multi-View Clustering (DIMVC).

**DMVC** focuses on effectively utilizing consistency and complementary information in multi-view data. Recent studies have advanced along three key aspects. 1) *View Representation Enhancement.* CoMVC (Trosten et al., 2021) proposes a simple method to avoid cluster inseparability caused by complete alignment; MFLVC (Xu et al., 2022) introduces a multi-level feature learning framework to resolve conflicts in multi-objective optimization; GCFAggMVC (Yan et al., 2023) strengthens sample structural relationships through global and cross-view feature aggregation. 2) *Semantic Consistency Improvement.* SEM (Xu et al., 2023) develops a self-weighting method to mitigate representation degradation caused by semantic inconsistency; CSOT (Zhang et al., 2024c) enhances multi-view semantic patterns through global semantic alignment; AECoDDC (Trosten et al., 2023) designs an end-to-end contrastive method incorporating DDC unsupervised loss (Trosten et al., 2021). 3) *Robustness Enhancement.* AR-DMVC and AR-DMVC-AM (Huang et al., 2024) addresses the vulnerability of DMVC models to adversarial attacks without explicitly defined attack targets.

**DIMVC** focus on scenarios with missing views (Fang et al., 2022). DSIMVC (Tang & Liu, 2022) proposes a dual-layer optimization framework that employs dynamic view completion and sample selection to mitigate performance degradation caused by semantically inconsistent view completion.

These methods demonstrate competitive performance on balanced multi-view data. However, in Section 5, we show that class imbalance significantly impacts both complete and incomplete multi-view clustering, substantially degrading clustering performance and model robustness.

### 2.2. Optimal transport and clustering

Optimal transport (OT) (Villani, 2009) has emerged as a powerful tool for distribution alignment in machine learning. While traditional OT faced computational challenges, the introduction of *entropy-regularized OT* (Cuturi, 2013) with the Sinkhorn algorithm enabled its practical applications. To address real-world scenarios with unequal mass

distributions, variants such as *partial OT* (Caffarelli & Mc-Cann, 2010) and *unbalanced OT* (Chizat et al., 2016) were developed.

The integration of OT in clustering has evolved from early *Wasserstein K-means* (Cuturi & Doucet, 2014) to deep learning methods (Luo et al., 2023; Lin et al., 2023a; Zhang et al., 2024b; Ben-Bouazza et al., 2022). SeLa (Asano et al., 2020) reformulates pseudo-label assignment as an OT problem. Recent methods addressing class imbalance have made significant progress. SLA (Tai et al., 2021) proposes class-proportional label assignment. DB-OT (Shi et al., 2024) develops dual-boundary constraints. $P^2OT$ (Zhang et al., 2024a) introduces progressive partial transport. These developments in OT, particularly its success in handling imbalanced data distributions, motivate our application of OT principles to *imbalanced multi-view clustering*.

## 3. Preliminary

In this section, we review the development of OT and lay the foundation for our subsequent imbalanced multi-view method. Given source distribution $r$ and target distribution $c$, their empirical distributions can be represented as:

$$r = \sum_{i=1}^{n} r_i \delta_{x_i}, \quad c = \sum_{j=1}^{m} c_j \delta_{y_j}, \tag{1}$$

where $\delta_{x_i}$ denotes the Dirac function, $x_i \in X$ and $y_j \in Y$ are samples from the source and target domains respectively.

Optimal transport theory aims to find an optimal transport plan $Q \in \mathbb{R}_+^{n \times m}$ that minimizes the transportation cost:

$$\min_{Q \in \Pi(r,c)} \langle Q, C \rangle, \tag{2}$$

where $C$ is the cost matrix, and the transport constraint set is defined as:

$$\Pi(r,c) = \{Q \in \mathbb{R}_+^{n \times m} \mid Q1_m = r, Q^\top 1_n = c\}, \tag{3}$$

where $1_m \in \mathbb{R}^{m \times 1}$, $1_n \in \mathbb{R}^{n \times 1}$ are all ones vector. To improve computational efficiency, Cuturi (Cuturi, 2013) introduced an entropic regularization term $H(Q)$, resulting in the regularized OT problem:

$$\min_{Q \in \Pi(r,c)} \langle Q, C \rangle + \epsilon H(Q). \tag{4}$$

This problem can be efficiently solved using the Sinkhorn algorithm. However, in practical applications, the masses of source and target distributions are often unequal. To address this, unbalanced OT (UOT) relaxes the marginal constraints through KL divergence:

$$\min_{Q \in \Pi(r,c)} \langle Q, C \rangle + \gamma_1 D_{\mathrm{KL}}(Q1_m \| r)$$
$$+ \gamma_2 D_{\mathrm{KL}}(Q^\top 1_n \| c), \tag{5}$$

where $\gamma_1$ and $\gamma_2$ control the degree of marginal constraint relaxation. Furthermore, partial OT (POT) introduces a total mass constraint $\lambda$ to achieve the transport of partial samples:

$$\min_{Q \in \Pi^\lambda(r,c)} \langle Q, C \rangle, \tag{6}$$

where $\Pi^\lambda(r,c) = \{Q \in \mathbb{R}_+^{n \times m} \mid Q1_m \leq r, Q^\top 1_n \leq c, 1_n^\top Q1_m = \lambda\}$.

We integrate UOT with POT to perceive the underlying class imbalance distribution in multi-view data.

## 4. Methodology

This section first elaborates on the importance of imbalanced multi-view clustering research and its specific problem setting. To address this problem, we propose PROTOCOL, a novel PaRtial Optimal TranspOrt-enhanced COntrastive Learning method, which consists of two key modules: multi-view POT label assignment (Section 4.2) and multi-view class-rebalanced contrastive learning (Section 4.3). Finally, we present a comprehensive overview of PROTOCOL and its training procedure in Section 4.4.

### 4.1. Motivation and Problem Setting

Despite significant progress in recent years, most existing DMVC methods still focus on potentially uniformly distributed datasets, which limits their practical applicability. In many real-world applications, datasets often exhibit imbalanced distributions, where the majority of data belongs to a few major classes while the rest is scattered across numerous minor classes. Thus, this paper investigates a more practical problem setting: imbalanced multi-view clustering, which faces two major challenges: *(i) how to perceive class imbalance distribution in an unsupervised scenario*; *(ii) how to mitigate representation degradation of minority samples.* These two challenges are tightly coupled: when the model perceives the class imbalance distribution of the data, it inevitably biases towards majority class samples, thereby leading to representation degradation of minority samples. The solutions to these two challenges are presented in Section 4.2 and Section 4.3, respectively.

**Problem Setting.** Given a multi-view dataset $\{x^v \in \mathbb{R}^{N \times D_v}\}_{v=1}^V$ with imbalance ratio $R$, which contains $N$ samples across $V$ views, where $x_i^v \in \mathbb{R}^{D_v}$ denotes the $D_v$-dimensional feature vector of the $i$-th sample from the $v$-th view. The imbalance ratio $R$ is defined as the ratio between the number of samples in the smallest and largest clusters, *i.e.*,

$$R = \min_k \{n_k\} / \max_k \{n_k\}, \tag{7}$$

where $n_k$ denotes the number of samples in the $k$-th cluster. The dataset contains $K$ imbalance clusters to be discovered.

## 4.2. Multi-view POT Label Allocation

We propose a multi-view POT label allocation method that learns imbalanced class distribution of multi-view data through multi-view representation learning and a POT-based self-labeling mechanism.

### 4.2.1. MULTI-VIEW REPRESENTATION LEARNING

First, the raw data $\boldsymbol{X}^v$ is transformed into latent representations $\boldsymbol{Z}^v = f_{\theta_v}(\boldsymbol{x}^v) \in \mathbb{R}^{N \times d}$ through a learnable encoder network $f_{\theta_v}$. To better capture the underlying cluster patterns, the representations are refined into structure-aware features $\boldsymbol{S}^v \in \mathbb{R}^{N \times d}$ by learning sample relationships through transformer attention mechanism (Vaswani et al., 2017). Finally, $\boldsymbol{S}^v$ from all views are aggregated to obtain the consensus representation $\boldsymbol{U} \in \mathbb{R}^{N \times d}$.

**Multi-view Prediction.** Given $\boldsymbol{U} \in \mathbb{R}^{N \times d}$, we first map it to logits through a cosine classifier $h : \mathbb{R}^d \to \mathbb{R}^K$. The predicted probability matrix $\boldsymbol{P} \in \mathbb{R}^{N \times K}$ is obtained by:

$$\boldsymbol{P} = \text{softmax}(h(\boldsymbol{U})). \tag{8}$$

Similarly, for each view $v \in \{1, \ldots, V\}$, we obtain view-specific predictions:

$$\boldsymbol{P}^v = \text{softmax}(h(\boldsymbol{Z}^v)). \tag{9}$$

### 4.2.2. SELF-LABELING MECHANISM WITH POT

Inspired by supervised learning with true labels $\boldsymbol{y}_i$, the model would be trained by minimizing the cross-entropy loss:

$$E(\boldsymbol{Q}|\boldsymbol{y}_1, \ldots, \boldsymbol{y}_N) = -\frac{1}{N} \sum_{i=1}^{N} \log \boldsymbol{Q}_{i,\boldsymbol{y}_i}. \tag{10}$$

where $\boldsymbol{Q}$ is the predicted label of the model. Studies have shown that supervised classification models can achieve superior performance with sufficient labeled data (Deng et al., 2009), which has led to extensive research on self-labeling methods (Lee et al., 2020; Zhang et al., 2023a; Li et al., 2023). Thus, we propose to incorporate self-labeling into multi-view clustering tasks. To address the requirement of true labels in Eq. (10), we introduce a posterior probability distribution $q(\boldsymbol{y}|\boldsymbol{x}_i)$ (denoted as matrix $\boldsymbol{T} \in \mathbb{R}^{N \times K}$, where $\boldsymbol{T}_{i,k}$ represents the probability of sample $i$ belonging to class $k$) and propose a multi-view learning based cross-entropy loss:

$$E(\hat{\boldsymbol{P}}, \boldsymbol{T}) = -\frac{1}{N} \sum_{i=1}^{N} \sum_{k=1}^{K} \boldsymbol{T}_{i,k} \log \hat{\boldsymbol{P}}_{i,k}, \tag{11}$$

where $\hat{\boldsymbol{P}}_{i,k} = \alpha \boldsymbol{P}_{i,k} + \frac{1-\alpha}{V} \sum_{v=1}^{V} \boldsymbol{P}_{i,k}^v$, in which $\boldsymbol{P}_{i,k}$ and $\boldsymbol{P}_{i,k}^v$ are obtained from Eqs. (8) and (9), respectively. Here

$\alpha \in [0,1]$ balances the importance between consensus and view-specific predictions.

**Imbalanced Multi-view Self-labeling.** To learn the imbalanced class distribution and avoid degenerate solutions, we add constraints that adaptively allocate labels to different clusters. Formally, the objective function for imbalanced multi-view self-labeling is:

$$\min_{\boldsymbol{T}, \hat{\boldsymbol{P}}} E(\hat{\boldsymbol{P}}, \boldsymbol{T}) \quad \text{s.t.} \quad \begin{cases} \boldsymbol{T}_{i,k} \in [0,1], & \forall i, k \\ \sum_{k=1}^{K} \boldsymbol{T}_{i,k} \leq 1, & \forall i \\ \sum_{i=1}^{N} \boldsymbol{T}_{i,k} \leq \lambda, & \forall k \end{cases}, \tag{12}$$

where $\lambda > 0$ is an adaptive parameter that adjusts the allocation for class $k$ to account for imbalanced distribution. The constraints ensure that each data point $x_i$ is assigned to exactly one label, while allowing the $N$ data points to be distributed among the $K$ classes in a way that reflects the class imbalances.

**Optimal Transport Formulation.** The above is an instance of combining *unbalanced OT* and *partial OT*. Let $\hat{\boldsymbol{P}}\frac{1}{N}$ be the joint probability distribution predicted by the model, and $\boldsymbol{T}\frac{1}{N}$ be the assigned joint probability distribution. Using the concept of regularized OT (Cuturi, 2013), $\boldsymbol{T}$ is relaxed as an element of the transportation polytope:

$$U(\boldsymbol{r}, \boldsymbol{c}) := \{\boldsymbol{T} \in \mathbb{R}_+^{N \times K} \mid \boldsymbol{T}\mathbf{1}_K = \boldsymbol{r}, \boldsymbol{T}^\top \mathbf{1}_N = \boldsymbol{c}\}, \tag{13}$$

where $\mathbf{1}_K$, $\mathbf{1}_N$ are all ones vector, so that $\boldsymbol{r}$ and $\boldsymbol{c}$ are the marginal projections of matrix $\boldsymbol{T}$ onto its rows and columns, respectively. In the imbalanced multi-view scene, we require $\boldsymbol{T}$ to be a matrix of conditional probability distribution that splits the data adaptively, which is captured by:

$$\boldsymbol{r} = \frac{1}{N} \cdot \mathbf{1}_N, \qquad \boldsymbol{c} = \frac{\lambda}{K} \cdot \mathbf{1}_K. \tag{14}$$

With this notation, using Eqs. (5) and (6), we can rewrite the objective function in Eq. (12) as:

$$\mathcal{L}_{\text{POT}} = \min_{\boldsymbol{T} \in U(\boldsymbol{r}, \boldsymbol{c})} \langle \boldsymbol{T}, -\log \hat{\boldsymbol{P}} \rangle_F + \beta D_{\text{KL}}(\boldsymbol{T}^\top \mathbf{1}_N \| \boldsymbol{c}) \tag{15}$$

where $U = \{\boldsymbol{T} \in \mathbb{R}_+^{N \times K} \mid \boldsymbol{T}\mathbf{1}_K \leq \boldsymbol{r}, \mathbf{1}_N^\top \boldsymbol{T}\mathbf{1}_K = \lambda\}$; here $\langle . \rangle$ is Frobenius dot-product, $\lambda$ is converted to the fraction of selected mass and will increase gradually, and $\beta$ is a scalar factor. The first term is exactly $\mathcal{L}$, the $D_{\text{KL}}$ term constrains cluster sizes, and the equality constraint ensures balanced sample importance.

### 4.2.3. PROGRESSIVE POT LABEL OPTIMIZATION

**Progressive POT Label Assignment.** The solution to the optimization problem in Eq. (15) yields the POT label matrix $\boldsymbol{T}$, where each entry $\boldsymbol{T}_{i,k}$ represents the probability of sample $i$ belonging to class $k$. The transport mass of POT

labels is regulated by $\lambda$, which progressively grows to enable the perception of class imbalance in multi-view data. Inspired by curriculum learning, when $\lambda$ is small in the early stages, only high-confidence samples from $\hat{P}$ are selected, minimizing the learning cost. As $\lambda$ increases, more samples participate in learning until the completion of difficult sample label assignment. This process naturally integrates imbalanced distribution perception through $\lambda$, eliminating the need for manual confidence thresholds.

Following (Tarvainen & Valpola, 2017; Laine, 2017; Zhang et al., 2024a), we update $\lambda$ by a sigmoid ramp-up function:

$$\lambda = \lambda_{\text{base}} + (\lambda_{\text{max}} - \lambda_{\text{base}}) \cdot e^{-5(1-\tau)^2}, \quad (16)$$

where $\lambda_{\text{base}}$ and $\lambda_{\text{max}}$ define the range of transported mass, and $\tau \in [0, 1]$ represents the normalized training progress.

**Efficient Scaling Solution.** The optimal transport plan, i.e., the POT label matrix $\boldsymbol{T}^*$, can be derived through an efficient scaling algorithm:

$$\boldsymbol{T}^* = \text{diag}(\boldsymbol{a})\boldsymbol{M}\text{diag}(\boldsymbol{b}), \quad (17)$$

where $\boldsymbol{M} = \exp(-\hat{\boldsymbol{P}}/\epsilon)$. $\boldsymbol{a}$ and $\boldsymbol{b}$ are two scaling coefficient vectors that can be obtained through the following recursion formula:

$$\boldsymbol{a} \leftarrow \frac{\boldsymbol{r}}{\boldsymbol{M}\boldsymbol{b}}, \quad \boldsymbol{b} \leftarrow \left(\frac{\boldsymbol{c}^*}{\boldsymbol{M}^\top \boldsymbol{a}}\right)^{\boldsymbol{f}}, \quad (18)$$

where $\boldsymbol{f} = \frac{\beta}{\beta+\epsilon}$. The recursion will continue until $\boldsymbol{b}$ converges. This solution elegantly combines the dynamic growth of $\lambda$ with efficient matrix scaling operations, enabling the model to progressively assign POT labels while considering multi-view consistency. The complete derivation can be found in Appendix A.

Through the learning of these components, PROTOCOL can effectively perceive the imbalanced distribution of multi-view data. This leads to the second challenge: *how to mitigate representation degradation of minority samples*, which will be addressed in Section 4.3.

### 4.3. Multi-view Class-rebalanced Contrastive Learning

While contrastive learning has become the de facto method in DMVC for learning semantically consistent representations, its performance significantly degrades under class-imbalanced settings. To analyze and address this limitation, we examine GCFAggMVC (Yan et al., 2023), a representative structure-guided contrastive loss:

$$\mathcal{L}_{\text{mvc}} = -\mathop{\mathbb{E}}_{\{\boldsymbol{H}_i^v, \boldsymbol{U}_1, \ldots, \boldsymbol{U}_N\}}\left[\log \frac{\mathcal{D}(\{\boldsymbol{H}_i^v, \boldsymbol{U}_i\})}{\sum\limits_{\substack{j=1 \\ j \neq i}}^{N} (1 - \boldsymbol{G}_{ij}^2)\mathcal{D}(\{\boldsymbol{H}_i^v, \boldsymbol{U}_j\})}\right], \quad (19)$$

where $\mathcal{D}(\{\boldsymbol{H}_i^v, \boldsymbol{U}_i\}) = \exp\left(\frac{\boldsymbol{H}_i^v \boldsymbol{U}_i}{\|\boldsymbol{H}_i^v\|\|\boldsymbol{U}_i\|} \cdot \frac{1}{\tau_f}\right)$ measures the similarity between the high-level feature $\boldsymbol{H}_i^v$ and the

concensus prototype $\boldsymbol{U}_i$, and $\boldsymbol{G}_{ij}$ represents the structural relationship between samples, with larger values indicating higher probability that samples belong to the same cluster.

In class-balanced scenarios, Eq. (19) effectively learns discriminative features as samples have sufficient opportunities to construct stable cross-view positive pairs through $\mathcal{D}(\{\boldsymbol{H}_i^v, \boldsymbol{U}_i\})$. The structural relationships $\boldsymbol{G}_{ij}$, learned through the transformer's self-attention mechanism, capture the inherent similarities between samples, enabling effective similarity-dissimilarity mining through $(1 - \boldsymbol{G}_{ij}^2)$.

However, when applied to imbalanced multi-view data, Eq. (19) leads to representation bias through two mechanisms. *First, from the sampling perspective,* majority samples dominate the mini-batch construction, resulting in more frequent positive pair formation through $\mathcal{D}(\{\boldsymbol{H}_i^v, \boldsymbol{U}_i\})$, while minority samples suffer from insufficient learning opportunities. *Second, from the optimization perspective,* although $\boldsymbol{G}_{ij}$ captures sample similarities through self-attention, the feature space becomes biased towards majority patterns due to their numerical advantage in contrastive learning, leading to compact clusters for majority samples but scattered distributions for minority ones.

To address the imbalanced multi-view representation learning problem, we propose a two-level rebalancing strategy that operates at both feature and class levels:

**Feature-level Rebalancing.** We introduce a logit-adjusted contrastive learning mechanism guided by POT labels:

$$\mathcal{L}_{\text{fea}}^{re} = -\mathop{\mathbb{E}}_{\{\boldsymbol{H}_i^v, \boldsymbol{U}_1, \ldots, \boldsymbol{U}_N\}}\left[\log \frac{\exp(\mathcal{D}(\{\boldsymbol{H}_i^v, \boldsymbol{U}_i\}) + \eta_i)}{\sum_{j=1}^{N} \exp(\mathcal{D}(\{\boldsymbol{H}_i^v, \boldsymbol{U}_j\}))}\right], \quad (20)$$

where $\eta_i = -\log p(T_i^*)$ is a logit-adjustment term, defined as the negative logarithm of the sample's cluster (pseudo label $T_i^*$) frequency. This term is small for high-frequency classes, causing minimal logit impact, and large for low-frequency classes, significantly adjusting logits to enhance the importance of under-represented patterns and compensate for representation bias.

**Class-level Rebalancing.** We propose a balanced class alignment strategy that introduces consensus prototypes $\boldsymbol{C} = \{c_1, c_2, \ldots, c_K\}$ from POT label. The class-level loss is formulated as:

$$\mathcal{L}_{\text{sem}}^{re} = \sum_{p_+ \in P_i^v \cup \boldsymbol{T}^*} -w(p_+) \log \frac{\exp(p_+ \cdot \psi(x_i))}{\sum_{p_k} \exp(p_k \cdot \psi(x_i))}, \quad (21)$$

where

$$w(p_+) = \begin{cases} w_v, & p_+ \in \boldsymbol{P}^v \\ w_t, & p_+ = \boldsymbol{T}^* \end{cases}. \quad (22)$$

Here $w(p_+)$ controls the contribution ratio between view-specific class $w_v$ and consensus class $w_t$. To make good use of contrastive learning and rebalance at the same time, we observe that $w_v = 0.8$ and $w_t = 0.2$ are a reasonable choice. The whole loss becomes closer to supervised cross-entropy.

In addition, the transformation function $\psi(\cdot)$ in Eq. (21) applies different strategies to view-specific predictions $\boldsymbol{P}^v$ and consensus prototypes $\boldsymbol{T}^*$:

$$p \cdot \psi(x_i) = \begin{cases} p \cdot \mathcal{G}(x_i), & p \in \boldsymbol{P}^v \\ p \cdot \mathcal{F}(x_i), & p \in \boldsymbol{T}^* \end{cases}, \quad (23)$$

where $\mathcal{G}(x_i)$ is an identity mapping, and $\mathcal{F}(x_i)$ computes class-frequency based $\boldsymbol{C}$ to address class imbalance:

$$\mathcal{F}(x_i) = x_i \cdot w_c, \quad w_c = \frac{n_c}{\sum_{k=1}^K n_k}. \quad (24)$$

Here, $w_c$ normalizes class contributions based on their relative frequencies in the training data, effectively rebalancing the influence of majority and minority classes during class alignment.

**Overall Objective.** The overall loss combines:

$$\mathcal{L}_{\mathrm{im}} = \mathcal{L}_{\mathrm{fea}}^{re} + \mathcal{L}_{\mathrm{sem}}^{re}. \quad (25)$$

Through the two-level rebalancing strategy, our method effectively addresses the class imbalance modeling challenge. At the *feature* level, logit-adjusted contrastive learning enhances the model's sensitivity to minority samples. At the *class* level, class-weighted consensus prototypes maintain global class consistency. This hierarchical design achieves balanced feature learning while preserving both instance discrimination and class structure, leading to robust representations for imbalanced multi-view data.

### 4.4. Overview and Training Strategy

PROTOCOL consists of three stages to learn multi-view representations:

**Stage 1: View-specific Representation.** We employ a reconstruction-based pre-training strategy to learn basic view-specific representation through auto-encoding. The reconstruction loss is formulated as:

$$\mathcal{L}_{\mathrm{rec}} = \sum_{v=1}^V \|\boldsymbol{x}^v - \hat{\boldsymbol{x}}^v\|_2^2, \quad (26)$$

where $\boldsymbol{x}^v$ and $\hat{\boldsymbol{x}}^v$ denote the input and reconstructed features of view $v$, respectively.

**Stage 2: Multi-view Representation Alignment.** After pre-training, we conduct structure-guided fine-tuning for cross-view alignment. The loss function combines feature

---

**Algorithm 1** Multi-view Imbalanced Learning Framework

**Input:** $\{\boldsymbol{x}^v\}_{v=1}^V$, $V$, $K$, maximum iterations $T_1, T_2, T_3$
**Output:** $\boldsymbol{T}^*$ and consensus representation $\boldsymbol{U}$
**Stage 1: View-specific Representation**
**for** $t = 1$ **to** $T_1$ **do**
  $\boldsymbol{Z}^v = f_{\theta_v}(\boldsymbol{x}^v)$   //    View-specific encoding
  $\hat{\boldsymbol{x}}^v = g_{\phi_v}(\boldsymbol{Z}^v)$   //    View-specific decoding
  Minimize $\mathcal{L}_{\mathrm{rec}} = \sum_{v=1}^V \|\boldsymbol{x}^v - \hat{\boldsymbol{x}}^v\|_2^2$
**end for**
**Stage 2: Multi-view Representation Alignment**
**for** $t = T_1$ **to** $T_2$ **do**
  $\boldsymbol{S}^v = \mathrm{Transformer}(\boldsymbol{Z}^v)$ // Structure-aware features
  $\boldsymbol{H}^v = \mathrm{MLP}(\boldsymbol{Z}^v)$   //    High level features
  $\boldsymbol{U} = \mathrm{Aggregate}(\{\boldsymbol{S}^v\}_{v=1}^V)$ // Consensus features
  $\boldsymbol{P}^v = \mathrm{softmax}(h(\boldsymbol{Z}^v))$ // View-specific predictions
  $\boldsymbol{P} = \mathrm{softmax}(h(\boldsymbol{U}))$ // Consensus predictions
  Minimize $\mathcal{L}_{\mathrm{align}} = \sum_{v=1}^V w_v(\mathcal{L}_{\mathrm{fea}}(\boldsymbol{H}^v, \boldsymbol{U}) + \mathcal{L}_{\mathrm{sem}}(\boldsymbol{P}^v, \boldsymbol{P}))$
**end for**
**Stage 3: Imbalanced Learning**
**for** $t = T_2$ **to** $T_3$ **do**
  Update $\lambda = \lambda_{\mathrm{base}} + (\lambda_{\mathrm{max}} - \lambda_{\mathrm{base}}) \cdot e^{-5(1-\tau)^2}$
  Compute $\boldsymbol{M} = \exp(-\hat{\boldsymbol{P}}/\epsilon)$
  Obtain POT labels via efficient scaling algorithm:
   $\boldsymbol{a} \leftarrow \frac{\boldsymbol{r}}{\boldsymbol{Mb}}, \boldsymbol{b} \leftarrow (\frac{\boldsymbol{c}^*}{\boldsymbol{M}^\top \boldsymbol{a}})^f$
   $\boldsymbol{T}^* = \mathrm{diag}(\boldsymbol{a})\boldsymbol{M}\mathrm{diag}(\boldsymbol{b})$
  Minimize $\mathcal{L}_{\mathrm{im}} = \mathcal{L}_{\mathrm{fea}}^{im} + \mathcal{L}_{\mathrm{sem}}^{im}$ // Rebalanced learning
**end for**

---

and class levels alignment:

$$\mathcal{L}_{\mathrm{align}} = \sum_{v=1}^V w_v(\mathcal{L}_{\mathrm{fea}}(\boldsymbol{H}^v, \boldsymbol{U}) + \mathcal{L}_{\mathrm{sem}}(\boldsymbol{P}^v, \boldsymbol{P})), \quad (27)$$

where $w_v$ is the adaptive weight for view $v$. The feature-level alignment loss $\mathcal{L}_{\mathrm{fea}}$ follows the structure-guided contrastive loss in Eq. (19). The class-level alignment loss $\mathcal{L}_{\mathrm{sem}}$ ensures consistency between view-specific semantic $\boldsymbol{P}^v$ and the common semantic $\boldsymbol{P}$ through:

$$\mathcal{L}_{\mathrm{sem}} = - \mathop{\mathbb{E}}_{\{\boldsymbol{P}_i^v, \boldsymbol{P}_1, \dots, \boldsymbol{P}_K\}} \left[ \log \frac{\mathcal{D}(\{\boldsymbol{P}_i^v, \boldsymbol{P}_i\})}{\sum\limits_{\substack{j=1 \\ j \neq i}}^K \mathcal{D}(\{\boldsymbol{P}_i^v, \boldsymbol{P}_j\})} \right]. \quad (28)$$

**Stage 3: Imbalanced Learning.** The final stage addresses the imbalanced learning challenge through POT-label-guided contrastive learning (See Sections 4.2 and 4.3).

These strategies enable PROTOCOL to handle imbalanced multi-view data, as summarized in Algorithm 1.

*Table 1.* Statistics of the multi-view datasets.

| Dataset | Samples | Classes | Views | View Features | Imbalance Ratio | | |
|---|---|---|---|---|---|---|---|
| | | | | | 0.1 $n_{min}/n_{max}$ | 0.5 $n_{min}/n_{max}$ | 0.9 $n_{min}/n_{max}$ |
| Hdigit (Chen et al., 2022) | 10000 | 10 | 2 | 784,256 | 100/1000 | 500/1000 | 900/1000 |
| Fashion (Xiao et al., 2017) | 10000 | 10 | 3 | 784,784,784 | 100/1000 | 500/1000 | 900/1000 |
| NUS-WIDE (Chua et al., 2009) | 5000 | 5 | 5 | 64,225,144,73,128 | 100/1000 | 500/1000 | 900/1000 |
| Caltech (Fei-Fei et al., 2004) | 1400 | 7 | 5 | 40,254,1984,512,928 | 20/200 | 100/200 | 180/200 |
| Cifar10 (Yan et al., 2023) | 50000 | 10 | 3 | 512,2048,1024 | 500/5000 | 2500/5000 | 4500/5000 |

*Table 2.* Performance Comparison with $R = 0.1$

| Methods | Hdigit | | | Fashion | | | Caltech | | | NUS-WIDE | | | Cifar10 | | |
|---|---|---|---|---|---|---|---|---|---|---|---|---|---|---|---|
| | ACC | NMI | PUR | ACC | NMI | PUR | ACC | NMI | PUR | ACC | NMI | PUR | ACC | NMI | PUR |
| DSIMVC | 0.754 | 0.790 | 0.902 | 0.699 | 0.685 | 0.755 | 0.521 | 0.430 | 0.656 | 0.132 | 0.192 | 0.677 | 0.768 | 0.751 | 0.864 |
| ARDMVC | 0.608 | 0.727 | 0.808 | 0.542 | 0.747 | 0.797 | 0.551 | 0.489 | 0.675 | 0.312 | 0.159 | 0.534 | 0.650 | 0.641 | 0.798 |
| ARDMVCAM | 0.629 | 0.759 | 0.824 | 0.538 | 0.751 | 0.799 | 0.569 | 0.521 | 0.702 | 0.336 | 0.185 | 0.613 | 0.564 | 0.646 | 0.748 |
| CoMVC | 0.708 | 0.833 | 0.894 | 0.535 | 0.706 | 0.756 | 0.618 | 0.618 | 0.701 | 0.423 | 0.206 | 0.619 | 0.590 | 0.601 | 0.747 |
| MFLVC | 0.696 | 0.682 | 0.828 | 0.804 | 0.865 | 0.919 | 0.640 | 0.596 | 0.656 | 0.437 | 0.309 | 0.786 | 0.846 | 0.905 | 0.926 |
| GCFAggMVC | 0.716 | 0.843 | 0.927 | 0.693 | 0.810 | 0.881 | 0.743 | 0.642 | 0.743 | 0.440 | 0.295 | 0.801 | 0.844 | 0.912 | 0.959 |
| AECoDDC | 0.598 | 0.682 | 0.783 | 0.788 | 0.855 | 0.912 | 0.370 | 0.226 | 0.476 | 0.395 | 0.174 | 0.590 | 0.722 | 0.833 | 0.911 |
| SEM | 0.825 | 0.835 | 0.947 | 0.768 | 0.851 | 0.915 | 0.727 | 0.642 | 0.727 | 0.417 | 0.296 | 0.797 | 0.851 | 0.918 | 0.965 |
| CSOT | 0.738 | 0.729 | 0.872 | 0.780 | 0.838 | 0.911 | 0.712 | 0.617 | 0.712 | 0.436 | 0.311 | 0.794 | 0.849 | 0.913 | 0.878 |
| PROTOCOL | **0.892** | **0.914** | **0.960** | **0.846** | **0.903** | **0.955** | **0.791** | **0.679** | **0.791** | **0.470** | **0.340** | **0.816** | **0.861** | **0.930** | **0.967** |

## 5. Experiment

### 5.1. Experimental Setup

To evaluate PROTOCOL, we establish a comprehensive benchmark on five widely-used multi-view datasets, as shown in Table 1. To simulate real-world imbalanced scenarios, we create imbalanced versions of these datasets with three imbalance ratios $R \in \{0.1, 0.5, 0.9\}$. The $R$ are kept consistent across different views. We place implementation details of PROTOCOL in the Appendix B. We compare PROTOCOL with nine state-of-the-art methods, including six DMVC methods (CoMVC (Trosten et al., 2021), MFLVC (Xu et al., 2022), AECoDDC (Trosten et al., 2023), GCFAggMVC (Yan et al., 2023), SEM (Xu et al., 2023), CSOT (Zhang et al., 2024c)), a DIMVC method (DSIMVC (Tang & Liu, 2022)), and two robust adversarial DMVC methods (AR-DMVC and AR-DMVC-AM (Huang et al., 2024)). We adopt three widely used metrics to evaluate clustering performance: Clustering Accuracy (ACC), Normalized Mutual Information (NMI), and Purity (PUR). Our code is available at https://github.com/Scarlett125/PROTOCOL.

### 5.2. Main Results

To simulate real-world scenarios, we evaluate all methods on imbalanced multi-view data. Experiments are conducted on five datasets under three imbalance ratios $R \in \{0.1, 0.5, 0.9\}$, as shown in Tables 2 to 4. Based on these results, we have the following observations.

**Impact of Class Imbalance.** The results reveal a consistent pattern: most baseline methods suffer from *performance degradation* under imbalanced scenarios. The degradation becomes more pronounced as $R$ decreases from 0.9 to 0.1, indicating that class imbalance poses a fundamental challenge to multi-view clustering.

**Superior Performance on Severe Imbalance.** In the most severe imbalance scenario ($R = 0.1$), while baseline methods suffer from substantial degradation, PROTOCOL shows remarkable robustness. It outperforms the second-best method by 6.7% ACC and 7.1% NMI on Hdigit, and by 4.8% ACC and 3.7% NMI on Caltech, validating its effectiveness in handling severe imbalance distribution.

**Robustness across Imbalance Ratios.** As the imbalance ratio increases ($R : 0.1 \rightarrow 0.5 \rightarrow 0.9$), most methods show improved performance. Notably, PROTOCOL maintains its superiority even under more balanced scenarios, achieving 0.987 ACC on Fashion ($R = 0.5$) and 0.993 ACC on Cifar10 ($R = 0.9$). These results demonstrate PROTOCOL's robust performance in handling both severely imbalanced data and varying levels of class imbalance distributions.

Furthermore, we train the models on training sets with three different imbalance ratios $R$ and evaluate their perception of different classes on balanced test sets. The results for Caltech and Hdigit are shown in Fig. 1, while those for Fashion and Cifar10 are provided in Appendix Fig. 5. All methods show improved performance on balanced test sets compared to imbalanced testing scenarios, mainly due to the reduced uncertainty from tail-class samples. Although the training imbalance ratio remains a key factor affecting performance, notably, PROTOCOL consistently achieves the best results across different $R$.

To further validate PROTOCOL's capability in handling tail-

*Table 3.* Performance Comparison with $R = 0.5$

| Methods | Hdigit | | | Fashion | | | Caltech | | | NUS-WIDE | | | Cifar10 | | |
|---|---|---|---|---|---|---|---|---|---|---|---|---|---|---|---|
| | ACC | NMI | PUR | ACC | NMI | PUR | ACC | NMI | PUR | ACC | NMI | PUR | ACC | NMI | PUR |
| DSIMVC | 0.955 | 0.894 | 0.958 | 0.727 | 0.714 | 0.732 | 0.529 | 0.439 | 0.551 | 0.149 | 0.175 | 0.545 | 0.905 | 0.804 | 0.905 |
| ARDMVC | 0.745 | 0.704 | 0.779 | 0.838 | 0.838 | 0.851 | 0.628 | 0.625 | 0.675 | 0.351 | 0.183 | 0.567 | 0.855 | 0.736 | 0.855 |
| ARDMVCAM | 0.879 | 0.932 | 0.928 | 0.844 | 0.850 | 0.859 | 0.657 | 0.639 | 0.693 | 0.375 | 0.208 | 0.621 | 0.972 | 0.951 | 0.972 |
| COMVC | 0.861 | 0.937 | 0.925 | 0.740 | 0.771 | 0.786 | 0.693 | 0.626 | 0.712 | 0.416 | 0.148 | 0.416 | 0.896 | 0.872 | 0.896 |
| MFLVC | 0.843 | 0.765 | 0.845 | 0.981 | 0.963 | 0.981 | 0.755 | 0.633 | 0.755 | 0.530 | 0.312 | 0.615 | 0.990 | 0.972 | 0.990 |
| GCFAggMVC | 0.979 | 0.944 | 0.979 | 0.980 | 0.957 | 0.980 | 0.761 | 0.638 | 0.760 | 0.525 | 0.299 | 0.619 | 0.989 | 0.970 | 0.989 |
| AECODDC | 0.843 | 0.895 | 0.910 | 0.975 | 0.952 | 0.975 | 0.451 | 0.304 | 0.495 | 0.426 | 0.164 | 0.458 | 0.877 | 0.802 | 0.877 |
| SEM | 0.980 | 0.945 | 0.980 | 0.982 | 0.961 | 0.979 | 0.759 | 0.611 | 0.707 | 0.475 | 0.261 | 0.565 | 0.990 | 0.971 | 0.982 |
| CSOT | 0.949 | 0.878 | 0.949 | 0.983 | 0.953 | 0.975 | 0.753 | 0.653 | 0.753 | 0.545 | 0.321 | 0.631 | 0.989 | 0.970 | 0.976 |
| PROTOCOL | **0.983** | **0.952** | **0.983** | **0.987** | **0.969** | **0.986** | **0.767** | **0.662** | **0.767** | **0.542** | **0.319** | **0.632** | **0.991** | **0.976** | **0.991** |

*Table 4.* Performance Comparison with $R = 0.9$

| Methods | Hdigit | | | Fashion | | | Caltech | | | NUS-WIDE | | | Cifar10 | | |
|---|---|---|---|---|---|---|---|---|---|---|---|---|---|---|---|
| | ACC | NMI | PUR | ACC | NMI | PUR | ACC | NMI | PUR | ACC | NMI | PUR | ACC | NMI | PUR |
| DSIMVC | 0.963 | 0.906 | 0.965 | 0.753 | 0.727 | 0.753 | 0.670 | 0.553 | 0.670 | 0.151 | 0.169 | 0.532 | 0.883 | 0.798 | 0.893 |
| ARDMVC | 0.859 | 0.831 | 0.859 | 0.871 | 0.885 | 0.880 | 0.660 | 0.592 | 0.677 | 0.396 | 0.216 | 0.598 | 0.875 | 0.753 | 0.875 |
| ARDMVCAM | 0.853 | 0.946 | 0.899 | 0.867 | 0.884 | 0.873 | 0.659 | 0.591 | 0.666 | 0.412 | 0.238 | 0.519 | 0.982 | 0.961 | 0.982 |
| COMVC | 0.906 | 0.930 | 0.906 | 0.808 | 0.808 | 0.816 | 0.640 | 0.580 | 0.646 | 0.383 | 0.097 | 0.383 | 0.963 | 0.941 | 0.963 |
| MFLVC | 0.913 | 0.820 | 0.913 | 0.989 | 0.967 | 0.989 | 0.802 | 0.715 | 0.802 | 0.614 | 0.351 | 0.614 | 0.991 | 0.976 | 0.991 |
| GCFAggMVC | 0.981 | 0.948 | 0.981 | 0.988 | 0.970 | 0.988 | 0.826 | 0.757 | 0.826 | 0.586 | 0.333 | 0.586 | 0.990 | 0.974 | 0.990 |
| AECODDC | 0.964 | 0.948 | 0.964 | 0.986 | 0.968 | 0.986 | 0.460 | 0.355 | 0.466 | 0.469 | 0.201 | 0.469 | 0.976 | 0.949 | 0.976 |
| SEM | 0.982 | 0.954 | 0.984 | 0.983 | 0.972 | 0.984 | 0.815 | 0.739 | 0.815 | 0.586 | 0.360 | 0.586 | 0.991 | 0.975 | 0.984 |
| CSOT | 0.952 | 0.888 | 0.952 | 0.982 | 0.970 | 0.982 | 0.815 | 0.721 | 0.815 | 0.556 | 0.282 | 0.556 | 0.990 | 0.977 | 0.982 |
| PROTOCOL | **0.984** | **0.956** | **0.984** | **0.991** | **0.976** | **0.990** | **0.896** | **0.830** | **0.878** | **0.634** | **0.367** | **0.634** | **0.993** | **0.981** | **0.993** |

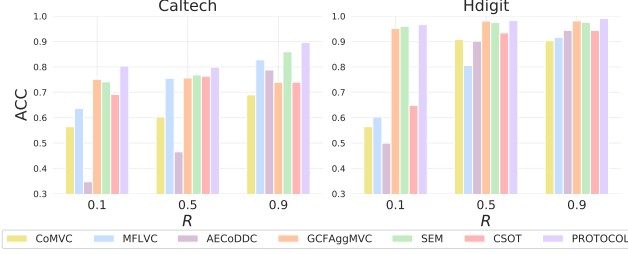

*Figure 1.* Clustering performance comparison on balanced test sets.

*Figure 2.* Head, Medium, and Tail comparison on several datasets.

class samples, we evaluate five competitive methods across head, medium, and tail classes on Caltech and NUS-WIDE datasets, with results for $R = 0.1$ shown in Fig. 2 (results for $R = 0.5$ are provided in Appendix Fig. 6). The results demonstrate that PROTOCOL not only maintains competitive performance on head classes but also improves the clustering effectiveness of medium and tail classes. This validates that *POT label assignment* and *class-rebalanced contrastive learning* effectively enhance the model's ability to learn discriminative features from tail classes, thereby addressing the representation bias caused by data scarcity.

To intuitively demonstrate the clustering effectiveness, we visualize the learned representations using t-SNE in Fig. 3. Compared with four baseline methods, PROTOCOL gener-

ates more compact and well-separated clusters. As shown in Fig. 3(e), the clusters learned by PROTOCOL exhibit clearer boundaries and more cohesive structures, indicating its superior ability under imbalanced scenarios.

### 5.3. Ablation Study

We conduct ablation studies on three datasets under $R \in \{0.1, 0.5\}$ to validate the effectiveness of each component in PROTOCOL. We compare three variants: (1) *Base*: Only includes reconstruction loss and multi-view consistency learning; (2) *w/ POT*: Incorporates POT-based label assignment; (3) *w/ POT+CLR* (denoted as *PROTOCOL*): Further adds POT-enhanced CLass-Rebalanced (CLR) contrastive learning. The results in Table 5 show the contribution of each

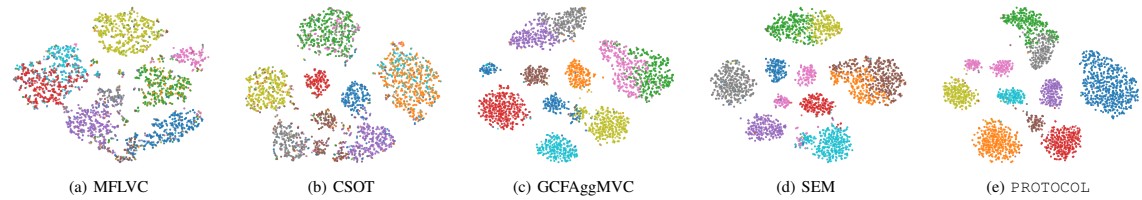

| (a) MFLVC | (b) CSOT | (c) GCFAggMVC | (d) SEM | (e) PROTOCOL |

*Figure 3.* The t-SNE visualization on Hdigit dataset.

*Table 5.* Ablation Study on Different Components

| Variants | Caltech | | | | Hdigit | | | | Fashion | | | |
|---|---|---|---|---|---|---|---|---|---|---|---|---|
| | $R = 0.1$ | | $R = 0.5$ | | $R = 0.1$ | | $R = 0.5$ | | $R = 0.1$ | | $R = 0.5$ | |
| | ACC | NMI | ACC | NMI | ACC | NMI | ACC | NMI | ACC | NMI | ACC | NMI |
| Base | 0.719 | 0.640 | 0.737 | 0.616 | 0.722 | 0.856 | 0.979 | 0.947 | 0.732 | 0.845 | 0.981 | 0.955 |
| w/ POT | 0.772 | 0.651 | 0.757 | 0.634 | 0.876 | 0.902 | 0.980 | 0.949 | 0.840 | 0.900 | 0.984 | 0.962 |
| w/ POT+CLR | **0.791** | **0.679** | **0.767** | **0.662** | **0.892** | **0.914** | **0.983** | **0.952** | **0.846** | **0.903** | **0.987** | **0.969** |

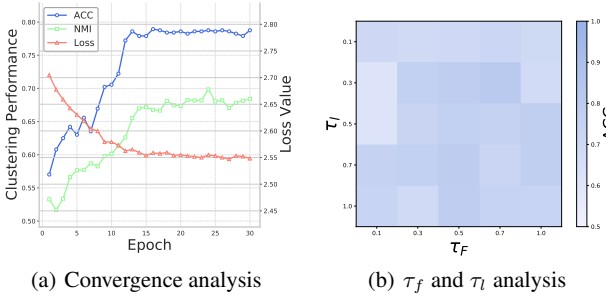

| (a) Convergence analysis | (b) $\tau_f$ and $\tau_l$ analysis |

*Figure 4.* Convergence and parameter sensitivity on Caltech.

component. First, incorporating POT-based label assignment (*w/ POT*) significantly outperforms the base model (*e.g.*, 15.4% ACC improvement on Hdigit when $R = 0.1$), validating the effectiveness of POT label assignment in handling class imbalance. Furthermore, adding POT-guided contrastive learning (*w/ POT+CLR*) further enhances the model performance, demonstrating its capability to improve tail-class representation. These results confirm that both components contribute to the overall performance: the POT mechanism provides the most significant improvement in handling class imbalance, while the class-rebalanced contrastive learning further optimizes tail-class representation.

### 5.4. Convergence and Parameter Sensitivity Analysis

We conduct convergence and parameter sensitivity analysis on the Caltech dataset, as shown in Fig. 4. As shown in Fig. 4(a), both ACC and NMI metrics show stable convergence trends, with the model achieving convergence after approximately 15 epochs and maintaining steady performance thereafter. In addition, we analyze two key hyperparameters: the feature and class temperature parameters $\tau_f$ and

$\tau_l$. As shown in Fig. 4(b), ACC fluctuates within a small range (0.7-0.8) as $\tau_f$ and $\tau_l$ vary, indicating the model's stability to temperature parameters. Based on these results, we set $\tau_f = 0.5$ and $\tau_l = 1.0$. The Appendix Fig. 7 shows similar stability for the semantic consistency parameter $a$ and the base learning weight $\lambda_{base}$, and we set $a = 0.5$, $\lambda_{base} = 0.1$.

## 6. Conclusion

In this paper, we propose PROTOCOL, a novel imbalanced multi-view clustering framework that effectively addresses the challenges of class imbalance and tail-class representation bias. Specifically, we introduced a POT-based label assignment mechanism that effectively perceives and handles class imbalance across multi-views, and designed a POT label-enhanced two-level class rebalanced contrastive learning strategy that enhances tail-class representation. Extensive experiments on five benchmark datasets demonstrate that PROTOCOL consistently outperforms existing methods, particularly in severely imbalanced scenarios. Ablation studies further validate the effectiveness of each component, where the POT mechanism provides significant improvements in handling class imbalance, while class-rebalanced contrastive learning further optimizes tail-class representation. These results indicate that PROTOCOL provides an effective solution for real-world multi-view clustering tasks where class imbalance is prevalent.

## Acknowledgements

This work was supported by National Natural Science Foundation of China (Nos.62176059, 62471148, and 62306075), China Postdoctoral Science Foundation and Fellowship Program of CPSF (No.2024278).

## Impact Statement

Our research on imbalanced multi-view clustering addresses a fundamental challenge in real-world scenarios, where data from multiple sources (*e.g.*, medical imaging modalities, multi-sensor recordings, and social media platforms) often exhibits natural class imbalance.

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

# A. Solving Process for POT Label

In this section, we detail how to solve the optimal transport by an efficient scaling algorithm. Our goal is to combine partial optimal transport problems and unbalanced optimal transport problems to perceive the class imbalance distribution in multi-view data, with the following objective function:

$$\mathcal{L}_{\text{POT}} = \min_{\boldsymbol{T} \in U(\boldsymbol{r}, \boldsymbol{c})} \langle \boldsymbol{T}, -\log \hat{\boldsymbol{P}} \rangle_F + D_{\text{KL}}(\boldsymbol{T}^\top \mathbf{1}_N \| \boldsymbol{c}) \tag{29}$$

$$\text{s.t.} \quad U = \{\boldsymbol{T} \in \mathbb{R}_+^{N \times K} \mid \boldsymbol{T} \mathbf{1}_K \leq \boldsymbol{r}, \mathbf{1}_N^\top \boldsymbol{T} \mathbf{1}_K = \lambda\}, \tag{30}$$

where $\boldsymbol{r} = \frac{1}{N} \cdot \mathbf{1}_N$ is the sample distribution constraint, $\boldsymbol{c} = \frac{\lambda}{K} \cdot \mathbf{1}_K$ is the class distribution constraint, so that $\boldsymbol{r}$ and $\boldsymbol{c}$ are the marginal projections of matrix $\boldsymbol{T}$ onto its rows and columns, respectively. $\lambda$ is the total mass constraint of the transport matrix $\boldsymbol{T}$, $\beta$ is a scalar factor.

Inspired by PU (Chapel et al., 2020) and P$^2$OT (Zhang et al., 2024a), we extend the original transport plan $\boldsymbol{T}$ by introducing a virtual cluster $\boldsymbol{\xi}$. Specifically, we denote the assignment of samples on the virtual cluster as $\boldsymbol{\xi}$. Then, we extend $\boldsymbol{T}$ to $\hat{\boldsymbol{T}}$, which satisfies the following constraints:

$$\hat{\boldsymbol{T}} = [\boldsymbol{T}, \boldsymbol{\xi}] \in \mathbb{R}^{N \times (K+1)}, \quad \boldsymbol{\xi} \in \mathbb{R}^{N \times 1}, \quad \hat{\boldsymbol{T}} \mathbf{1}_{K+1} = \frac{1}{N} \mathbf{1}_N. \tag{31}$$

Due to $\mathbf{1}_N^\top \boldsymbol{T} \mathbf{1}_K = \lambda$, we know that:

$$\mathbf{1}_N^\top \boldsymbol{\xi} = 1 - \lambda. \tag{32}$$

Therefore,

$$\hat{\boldsymbol{T}}^\top \mathbf{1}_N = \begin{bmatrix} \boldsymbol{T}^\top \mathbf{1}_N \\ \boldsymbol{\xi}^\top \mathbf{1}_N \end{bmatrix} = \begin{bmatrix} \boldsymbol{T}^\top \mathbf{1}_N \\ 1 - \lambda \end{bmatrix}. \tag{33}$$

We denote $\boldsymbol{C} = [-\log \boldsymbol{P}, \mathbf{0}_N]$ and replace $\boldsymbol{T}$ with $\hat{\boldsymbol{T}}$, thus the Eq. (31) can be rewritten as follows:

$$\min_{\hat{\boldsymbol{T}} \in U} \langle \hat{\boldsymbol{T}}, \boldsymbol{C} \rangle_F + D_{\text{KL}}(\hat{\boldsymbol{T}}^\top \mathbf{1}_N \| \boldsymbol{c}^*; \boldsymbol{\beta}) \tag{34}$$

$$\text{s.t.} \quad U = \{\hat{\boldsymbol{T}} \in \mathbb{R}_+^{N \times (K+1)} \mid \hat{\boldsymbol{T}} \mathbf{1}_{K+1} = \frac{1}{N} \mathbf{1}_N\}, \quad \boldsymbol{c}^* = \begin{bmatrix} \frac{\lambda}{K} \mathbf{1}_K \\ 1 - \lambda \end{bmatrix}, \quad \boldsymbol{\beta}_{K+1} \to +\infty. \tag{35}$$

Note that we introduce the weighted KL divergence $D_{\text{KL}}^\beta$ with $\boldsymbol{\beta}_{K+1} \to +\infty$ to strictly enforce the virtual cluster constraint. The sample constraint changes from inequality ($\leq$) to equality ($=$) because the virtual cluster $\boldsymbol{\xi}$ absorbs the remaining mass, ensuring that the total mass of each sample exactly equals $\frac{1}{N}$. The weighted KL divergence is defined as:

$$D_{\text{KL}}(\boldsymbol{x} \| \boldsymbol{y}; \boldsymbol{w}) = \sum_i w_i x_i \log \frac{x_i}{y_i}. \tag{36}$$

Following Cuturi (Cuturi, 2013), we introduce an entropy regularization term to solve the optimal transport efficiently. The regularized version becomes:

$$\min_{\hat{\boldsymbol{T}} \in U} \langle \hat{\boldsymbol{T}}, \boldsymbol{C} \rangle_F - \epsilon H(\hat{\boldsymbol{T}}) + D_{\text{KL}}(\hat{\boldsymbol{T}}^\top \mathbf{1}_N \| \boldsymbol{c}^*; \boldsymbol{\beta}). \tag{37}$$

Using the properties of entropy regularization:

$$\langle \hat{\boldsymbol{T}}, \boldsymbol{C} \rangle_F - \epsilon H(\hat{\boldsymbol{T}}) = \epsilon \langle \hat{\boldsymbol{T}}, \boldsymbol{C}/\epsilon + \log \hat{\boldsymbol{T}} \rangle_F \tag{38}$$

$$= \epsilon \langle \hat{\boldsymbol{T}}, \log \frac{\hat{\boldsymbol{T}}}{\exp(-\boldsymbol{C}/\epsilon)} \rangle_F \tag{39}$$

$$= D_{\text{KL}}(\hat{\boldsymbol{T}} \| \exp(-\boldsymbol{C}/\epsilon); \epsilon \mathbf{1}_{N \times (K+1)}). \tag{40}$$

---

**Algorithm 2** Efficient Scaling Algorithm for POT

---

**Require:** Cost matrix $P$, mass $\lambda$, regularization $\epsilon$, weights $\boldsymbol{\beta}$
1: $\boldsymbol{C} \leftarrow [-\log \boldsymbol{P}, \boldsymbol{0}_N]$
2: $\boldsymbol{M} \leftarrow \exp(-\boldsymbol{C}/\epsilon)$
3: $\boldsymbol{c}^* \leftarrow [\frac{\lambda}{K}\boldsymbol{1}_K; 1-\lambda]$
4: $\boldsymbol{b} \leftarrow \boldsymbol{1}_{K+1}$
5: $f_j \leftarrow \frac{\beta_j}{\beta_j+\epsilon}$ for $j = 1,\ldots, K+1$
6: **while** not converge **do**
7: $\quad \boldsymbol{a} \leftarrow \frac{\frac{1}{N}\boldsymbol{1}_N}{\boldsymbol{M}\boldsymbol{b}}$
8: $\quad \boldsymbol{b} \leftarrow (\frac{\boldsymbol{c}^*}{\boldsymbol{M}^\top \boldsymbol{a}})^f$
9: **end while**
10: $\boldsymbol{T}^* \leftarrow \text{diag}(\boldsymbol{a})\boldsymbol{M}\text{diag}(\boldsymbol{b})$
11: **return** First $K$ columns of $\boldsymbol{T}^*$ as $\boldsymbol{T}$

---

Let $\boldsymbol{M} = \exp(-\boldsymbol{C}/\epsilon)$, the problem becomes:

$$\min_{\hat{\boldsymbol{T}} \in U} D_{\text{KL}}(\hat{\boldsymbol{T}}\|\boldsymbol{M}) + D_{\text{KL}}(\hat{\boldsymbol{T}}^\top \boldsymbol{1}_N \|\boldsymbol{c}^*; \boldsymbol{\beta}). \tag{41}$$

Taking the derivative with respect to $\hat{\boldsymbol{T}}_{ij}$ and setting it to zero:

$$\frac{\partial}{\partial \hat{\boldsymbol{T}}_{ij}}[D_{\text{KL}}(\hat{\boldsymbol{T}}\|\boldsymbol{M}) + D_{\text{KL}}(\hat{\boldsymbol{T}}^\top \boldsymbol{1}_N\|\boldsymbol{c}^*; \boldsymbol{\beta})] \tag{42}$$

$$= \epsilon(\log \hat{\boldsymbol{T}}_{ij} - \log \boldsymbol{M}_{ij} + 1) + \beta_j \log \frac{[\hat{\boldsymbol{T}}^\top \boldsymbol{1}_N]_j}{c_j^*} = 0. \tag{43}$$

This leads to:

$$\hat{\boldsymbol{T}}_{ij} = \boldsymbol{M}_{ij} \exp(-1 - \frac{\beta_j}{\epsilon} \log \frac{[\hat{\boldsymbol{T}}^\top \boldsymbol{1}_N]_j}{c_j^*}) \tag{44}$$

$$= \boldsymbol{M}_{ij} \exp(-1)(\frac{c_j^*}{[\hat{\boldsymbol{T}}^\top \boldsymbol{1}_N]_j})^{\frac{\beta_j}{\epsilon}}. \tag{45}$$

Let $f_j = \frac{\beta_j}{\beta_j+\epsilon}$, then $\frac{\beta_j}{\epsilon} = \frac{f_j}{1-f_j}$. After absorbing constants into the scaling factors:

$$\hat{\boldsymbol{T}}_{ij} = a_i \boldsymbol{M}_{ij} b_j. \tag{46}$$

This leads to the efficient scaling algorithm iterative updates:

$$\boldsymbol{a} \leftarrow \frac{\frac{1}{N}\boldsymbol{1}_N}{\boldsymbol{M}\boldsymbol{b}}, \tag{47}$$

$$\boldsymbol{b} \leftarrow (\frac{\boldsymbol{c}^*}{\boldsymbol{M}^\top \boldsymbol{a}})^f. \tag{48}$$

The final transport plan can be recovered as:

$$\boldsymbol{T}^* = \text{diag}(\boldsymbol{a})\boldsymbol{M}\text{diag}(\boldsymbol{b}). \tag{49}$$

Based on the above derivation, we summarize our solution in Algorithm 2.

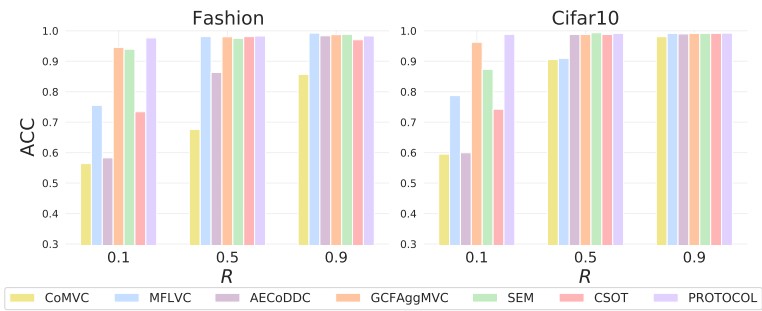

*Figure 5.* Clustering performance comparison on balanced test sets.

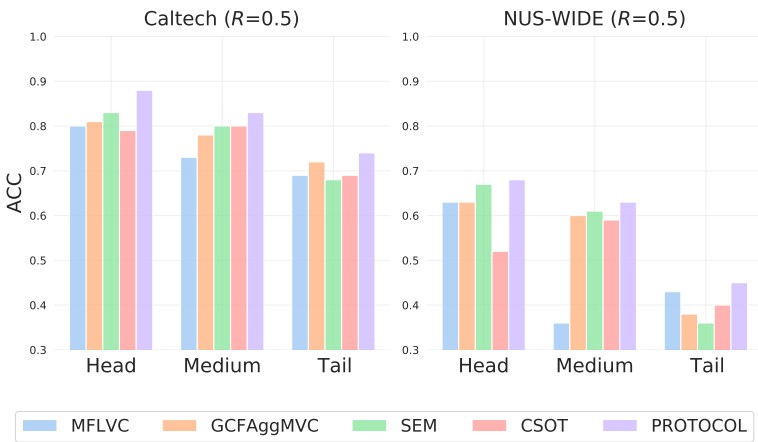

*Figure 6.* Head, Medium, and Tail comparison on several datasets.

## B. Experimental Supplements

**Implementation Details**    For network architecture, we employ four-layer MLPs for both encoder and decoder. Specifically, the encoder architecture consists of layers with dimensions (input_dim, 500, 500, 2000, 512), while the decoder follows a symmetric structure with dimensions (512, 2000, 500, 500, input_dim). Each layer is followed by ReLU activation except for the output layer. The high-level features are obtained by projecting the 512-dimensional features to 128 dimensions through an additional MLP. The optimization is performed using Adam optimizer with learning rate 1e-3. The training process consists of three stages: view-specific feature learning with 200 epochs for reconstruction loss, consensus learning with 50-100 epochs for multi-view consistency, and class-rebalanced enhancement with 50-100 epochs for imbalance learning. We set the batch size to 256. We report the average results over three runs with different random seeds.

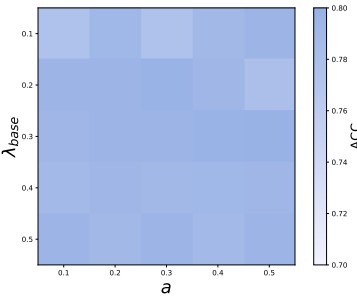

*Figure 7.* Parameter sensitivity analysis on Caltech dataset.

