# OpenReview forum: "PROTOCOL: Partial Optimal Transport-enhanced Contrastive Learning for Imbalanced Multi-view Clustering"
_ICML.cc/2025/Conference — ICML 2025 poster_

### Official Review · Reviewer_mKm8 · 2025-03-12

**Overall Recommendation:** 2

**Summary:**

The paper addresses the class imbalance issue in multi-view clustering by combining UOT and POT to perceive class imbalance, and uses POT-enhanced class rebalance to mitigate the representation degradation of minority samples in contrastive learning. Through comparisons across multiple datasets and multi-view clustering algorithms, the paper demonstrates the superiority of the proposed method under different imbalance ratios.

**Claims And Evidence:**

No. The paper proposes achieving adaptive perception of class imbalance through the adjustment of lambda. However, the explanation does not provide readers with a clear understanding of the adaptive adjustment process of lambda.

**Essential References Not Discussed:**

The core of the paper lies in introducing Optimal Transport into multi-view clustering. Although the paper provides some transitions and background, the process is not clearly articulated, making it difficult to understand.

**Experimental Designs Or Analyses:**

The study conducts experimental comparisons based on five commonly used multi-view datasets. However, the experimental setup does not clearly explain how the class-imbalanced datasets were constructed from these five datasets for the comparative experiments in Table 2. Additionally, the experimental procedures for Figures 1 and 2 are not clearly described.

**Methods And Evaluation Criteria:**

The comparative experimental results and related analysis demonstrate the superiority of the proposed method in addressing the class imbalance problem. However, the explanation of the proposed method is difficult to understand.

**Other Comments Or Suggestions:**

[1]Given that the paper does not provide an easily understandable explanation of the proposed method and that data preprocessing is required to evaluate its clustering performance on class-imbalanced multi-view data, this process should be explicitly presented. Therefore, anonymized open-source code should be considered to facilitate a comprehensive understanding and verification of the proposed method’s effectiveness.

**Other Strengths And Weaknesses:**

Strength:
The authors keenly identified the class imbalance problem in multi-view clustering and addressed it by introducing Optimal Transport, tackling the issue from two perspectives: perceiving class imbalance and mitigating representation degradation of minority samples.
Weakness:
1) he paper does not provide a clear and intuitive explanation of how Optimal Transport addresses class imbalance, making it difficult for readers to comprehend the construction of the proposed method.
2) The authors conducted experiments with different imbalance ratios on common multi-view datasets, but the specific preprocessing steps were not detailed.
3) Furthermore, the experimental procedures for Tables 1 and 2 were not provided.

**Questions For Authors:**

[1]Why do the constraint conditions in Equation (12) reflect the distribution pattern of class imbalance?

**Relation To Broader Scientific Literature:**

The primary contribution of this study lies in addressing the class imbalance problem in multi-view clustering by optimizing it with partial optimal transport. Optimal transport has demonstrated strong capability in handling class imbalance problems in previous studies. Based on this, the authors introduce it into multi-view clustering, aiming to mitigate class imbalance within this context.

**Theoretical Claims:**

The main text does not include proofs for the theoretical claims.

---

> ### Author Rebuttal · Authors · 2025-03-31
>
> We sincerely appreciate your recognition of our work's motivation and method.We are also deeply grateful for your thorough review and valuable suggestions.
>
> **Q1:** A intuitive explanation of how Optimal Transport addresses class imbalance.
>
> **A1:** Thanks for your suggestion. We would like to draw your attention to **Subsection 4.1.3**. Specifically, our method perceives class imbalance distributions through the following process:
>
> Given the model's label predictions $\hat{\mathbf{P}}$ and prior distribution constraints $U$, PROTOCOL dynamically adjusts the transport mass (via $\lambda$) to gradually assign samples to imbalanced clusters (POT label matrix $\mathbf{T}$):
>
> (1) First assigns high-confidence samples (low transport costs);
>
> (2) Gradually incorporates lower-confidence samples as transport mass increases;
>
> (3) Naturally forms labels reflecting true class distribution.
>
> Combines **imbalanced labels** with **class-rebalanced learning (Section 4.2)** to address minority class representation.
>
> We will enhance the logical connections between modules in the **Methodology** section of our revised version.
>
> **Q2:** Specific preprocessing.
>
> **A2:**  Thank you for your helpful suggestion. The data preprocessing steps are as follows:
>
> step1: We start with the class-balanced datasets.
>
> step2: Based on the imbalance ratio $R$ as defined in Eq. (7), we calculate the sample size for each class in descending order. For instance, with Hdigit dataset at $R$=0.1, class 1 retains all 1000 samples while class 10 keeps 100 samples. The intermediate classes' sample sizes grow linearly. Samples: {1000, 774, 599, 464, 359, 278, 215, 167, 129, 100}.
>
> step3: For each class, we ensure that the same sample indices are maintained across all views.
>
> step4: To ensure stability, we employ fixed random seeds during sample selection.
>
> This preprocessing transforms the dataset into an imbalanced class distribution. The code will be made publicly available upon acceptance.
>
> **Q3:** The experimental procedures for Tables 1 and 2.
>
> **A3:** Thank you for your suggestion. Since you also mentioned Figure 1 and 2, we explain them together.
> * Tables 1 and 2: PROTOCOL's implementation is given in Appendix B. Dataset and preprocessing are provided in Table 4 and A2, respectively.
> * Figures 1 and 2:
>     * Figure 1: The model is trained on imbalanced training sets and evaluated on balanced test sets to assess its perception of different classes. The results demonstrate PROTOCOL's robustness.
>     * Figure 2: We categorize all classes into three groups: Head, Medium, and Tail. The distribution varies by class size:
>         * For 10-class: Head (first 3 classes), Medium (middle 4 classes), Tail (last 3 classes).
>         * For 7-class: Head (2), Medium (3), Tail (2).
>         * For 5-class: Head (1), Medium (3), Tail (1).
> This categorization aligns with the Head-Medium-Tail definition, as majority samples fall into head classes while minority samples belong to tail classes. The results demonstrate our method's superiority, validating its effectiveness in perceiving actual class distributions.
>
> **Q4:** An easy explanation of the proposed method, data preprocessing, and the anonymous code for validation.
>
> **A4:** Thank you for your suggestion.
>
> * **About an easily explanation of PROTOCOL:** See A1.
>
> * **Data preprocessing:** See A2.
>
> * **Anonymous code:** We have provided our code through anonymous link (https://zenodo.org/records/15119555).
>
>   * **For Testing:** We provide a pre-trained model on the Hdigit dataset with $R$=0.1 to help verify our method's effectiveness.
>
>   * **For Training:** We have released network.py and train.py, which demonstrate the training pipeline to facilitate understanding of our framework.
>
>   * **Environment Setup:** Please create a virtual environment following our instructions for smooth execution of our code.
>
> The complete source code will be made publicly available upon paper acceptance.
>
> **Q5:** Analysis that Eq. (12)'s constraints reflect class imbalance distribution.
>
> **A5:** Thank you for your suggestion. Eq. (12) captures the class imbalance distribution pattern through two key mechanisms:
>
> (1) The constraints  $\sum_{k=1}^K \mathbf{T}_{i,k} \leq 1$ implement a soft label assignment mechanism, enabling flexible sample-to-class assignments. This allows samples to have varying degrees of association with different classes.
>
> (2) The constraint $\sum_{i=1}^N \mathbf{T}_{i,k} \leq \lambda$ introduces an adaptive parameter $\lambda$ to regulate the maximum mass for each class. During the dynamic adjustment of  $\lambda$, higher confidence samples (majority classes) receive larger mass assignments, while lower confidence samples (minority classes) receive smaller mass assignments.
>
> Then, we transform Eq. (12) into Eq. (15), enabling the model to adaptively capture the inherent patterns of imbalanced class distributions.

---

### Official Review · Reviewer_jNCt · 2025-03-13

**Overall Recommendation:** 3

**Summary:**

In this paper, a novel Partial Optimal Transmission (POT) enhanced contrast learning framework, PROTOCOL, is proposed to address the class imbalance challenge in multi-view clustering. A two-level rebalancing strategy achieves balanced feature learning as well as consistency in view-specific and view-sharing allocation.

**Claims And Evidence:**

Yes

**Essential References Not Discussed:**

No

**Experimental Designs Or Analyses:**

Yes

**Methods And Evaluation Criteria:**

Yes

**Other Comments Or Suggestions:**

no

**Other Strengths And Weaknesses:**

Strengths:
- Good representation.
- Comprehensive experiments: The superiority and robustness of the method are verified on multiple datasets, covering different imbalance ratios, and ablation studies and visualization analysis are performed.

Weaknesses:
- Is w(p+) set empirically? Why manually specify such a ratio between view-specific and consensus class alignment? Why not let the model learn adaptively.

**Questions For Authors:**

- How to aggregate view-specific representations into consensus representation U?
- In Figure 1, why is there still an imbalanced ratio on the balanced test set?
- It may be clearer for the authors to show a comparison of the effects of different methods' visualizations on synthetic extreme imbalance dataset. Since the visualization results from Fig. 3 show that most of the methods can identify those small clusters, such a demonstration may not achieve the authors' original intention.

**Relation To Broader Scientific Literature:**

Unbalanced multi-view data is very common in real-world scenarios, but has not been explored much. This paper achieves balanced learning by modifying the paradigm of contrastive learning to make the model more sensitive to minority samples.

**Theoretical Claims:**

Yes

---

> ### Author Rebuttal · Authors · 2025-03-31
>
> We sincerely appreciate your recognition of both the novelty of our method and the practical value of our motivation, as well as your positive feedback on our paper's representation and experiments. We are also deeply grateful for your thorough review and valuable suggestions.
>
> **Q1:** The empirical setting of w(p+) and the possibility of adaptive learning by the model.
>
> **A1:** Thank you for your helpful suggestion.
>
> Yes, we empirically set $w(p_+)$ to 0.8 and 0.2 for the two settings based on experimental validation.
>
> Following your suggestion, we implemented adaptive learning for $w(p_+)$ and validated it on three datasets with $R$=0.1. The results show performance improvements of 0.3%\~0.5% compared to fixed parameters. Notably, the learned parameters (ranging from 0.664\~0.805 and 0.195\~0.336) align well with our empirical values (0.8/0.2), validating our empirical setting. We implemented this by randomly initializing the two settings in [0,1], with their sum constrained to 1.
>
> |         | ACC (Fixed: 0.8/0.2) |   ACC (Adaptive)    | $w(p_+)$ (Learned) |
> | :-----: | :----------------------: | :---------------------: | :---------------------: |
> | Caltech |          0.791           | 0.796 ($\uparrow$ 0.05) |       0.805/0.195       |
> | Hdigit  |          0.892           | 0.895 ($\uparrow$ 0.03) |       0.664/0.336       |
> | CIFAR10 |          0.861           | 0.864 ($\uparrow$ 0.03) |       0.782/0.218       |
>
> Given the improved performance and greater flexibility of adaptive learning, we will adopt this improvement in the revised version. We again thank you for the constructive suggestion.
>
> **Q2:**  Aggregation of view-specific representations into consensus representation.
>
> **A2:**  Thank you for your comment. The consensus representation  $\mathbf{U}$ is obtained through the following steps:
>
> step1: View-specific representations $\mathbf{Z}^v$ are learned through autoencoders from original data.
>
> step2: Inter-sample structural relationships are captured in relationship matrix $\mathbf{G}$ through a Transformer-based self-attention mechanism.
>
> step3: Structure-aware representations are computed as $\mathbf{S}^v=\mathbf{Z}^v\mathbf{G}$ for each view.
>
> step4: View weights $\mathbf{w}^v$ are learned through a view weight learning module.
>
> step5: Final consensus representation is obtained by weighted fusion: $\mathbf{U}= \sum_{v=1}^{V}\mathbf{w}^v\mathbf{S}^v$.
>
>
> **Q3:** Regarding the imbalanced ratio $R$ in Figure 1.
>
> **A3:** Thank you for your comment. The imbalance ratio $R$ only applies to the training set, while the test set remains balanced. PROTOCOL maintains superior performance across different train imbalance ratios, demonstrating its effectiveness and robustness in handling class-imbalanced multi-view data.
>
> **Q4:** Adding visualization results for more extreme imbalance data.
>
> **A4:** Thank you for your insightful suggestion. Following your recommendation,  we conducted tests on the Hdigit dataset with an even more extreme imbalance ratio of $R$=0.05, with visualization results shown in **Figure A3** of the PDF file provided in the anonymous link (https://zenodo.org/records/15117646). The results demonstrate that, compared to baseline methods, PROTOCOL can effectively identify smaller clusters and clearly distinguish cluster structures of varying scales, validating our method's effectiveness and robustness under extreme imbalance ratios.
>
> To more intuitively demonstrate PROTOCOL's ability to perceive imbalanced data distributions, we conducted a quantitative analysis of the clustering results from Figure 3 in the original paper and **Figure A3**, where we calculated the number of samples in each class from the test results and computed the actual imbalance ratios.
>
> As shown in the table below, when the imbalance ratio $R$=0.1, other methods produced actual imbalance ratios between **0.26~0.38**, while PROTOCOL achieved an actual imbalance ratio of only **0.14**. Similarity, when the imbalance ratio $R$=0.05, other methods produced actual imbalance ratios between **0.23~0.37**, while PROTOCOL achieved an actual imbalance ratio of only **0.12**. This indicates that our method can more accurately perceive and maintain the class distribution characteristics of the original data.
>
> | Actual_$R$ | MFLVC | CSOT | GCFAggMVC | SEM  | PROTOCOL |
> | :----: | :---: | :--: | :-------: | :--: | :------: |
> | $R$=0.1  | 0.26  | 0.28 |   0.39    | 0.38 | **0.14** |
> | $R$=0.05 | 0.28  | 0.25 |   0.23    | 0.37 | **0.12** |

---

> > ### Comment · Reviewer_jNCt · 2025-04-04
> >
> > I appreciate the answers and clarification. I have no concerns about the work and hence keep the rating.

---

> > > ### Author Response · Authors · 2025-04-07
> > >
> > > Thank you for your positive assessment of our work. We sincerely appreciate your time and effort.

---

### Official Review · Reviewer_sJkj · 2025-03-13

**Overall Recommendation:** 3

**Summary:**

The paper introduces PROTOCOL, a new method for imbalanced multi-view clustering. It combines partial optimal transport (POT) with contrastive learning. The approach solves two main problems: perceiving class imbalance distributions through POT-based label assignment and reducing the representation degradation of minority samples using rebalancing strategies at the feature and class levels. Tests on multiple datasets show that PROTOCOL performs better, especially when data is highly imbalanced.

**Claims And Evidence:**

The claims presented in the paper are robustly bolstered by the experimental evidence.
1.The claims about POT’s effectiveness for imbalanced MVC are supported by ablation studies and t-SNE visualizations, showing clearer cluster boundaries for imbalanced multi-view clustering.
2.The superiority over baselines is validated across all datasets and metrics.

**Essential References Not Discussed:**

The literature research section of the paper is quite substantial, but there are still some related papers that have not been mentioned.
Including recent single-view and multi-view class imbalance approaches (e.g., [1]) would enhance the paper's comprehensiveness and contextualize its contributions more effectively.
[1] Zhou Q, Sun B. Adaptive K-means clustering based under-sampling methods to solve the class imbalance problem[J]. Data and Information Management, 2024, 8(3): 100064.

**Experimental Designs Or Analyses:**

The work performs several experiments on 5 datasets with three different imblance ratios

**Methods And Evaluation Criteria:**

Yes

**Other Comments Or Suggestions:**

Refer to the weakness

**Other Strengths And Weaknesses:**

Strengths:
1. Originality: The novel integration of POT and contrastive learning offers a fresh approach to imbalanced multi-view clustering.
2. Practical Value: Addressing real-world imbalanced data challenges highlights the paper’s potential impact in applications like ecological monitoring.
3. Thorough Evaluation: Rigorous experiments across datasets and imbalance scenarios demonstrate the framework’s performance.
Weaknesses:
1. The computational cost of POT might pose challenges for large-scale applications, which could be a direction for future optimization.
2. The paper's structure could be clarified to better highlight the logical connections between components. A more explicit explanation of how each module addresses specific challenges would help readers appreciate the framework's coherence.

**Questions For Authors:**

1. Theoretical Justification: Could the authors elaborate on the theoretical foundation of the POT scaling algorithm, such as its convergence properties? This would significantly enhance the paper's theoretical contributions.
2. Runtime Analysis: How does the computational cost of PROTOCOL scale with dataset size?

**Relation To Broader Scientific Literature:**

The paper introduces a new method for imbalanced multi-view clustering by combining partial optimal transport (POT) with contrastive learning.

**Theoretical Claims:**

Yes. The paper is based on theoretical concepts of OT and contrastive learning.

---

> ### Author Rebuttal · Authors · 2025-03-31
>
> We sincerely appreciate your recognition of our work's novelty and its potential impact in enhancing multi-view clustering for real-world imbalanced scenarios, as well as your positive feedback on our experimental results. We are also deeply grateful for your thorough review and helpful suggestions.
>
> **Q1:** The computational cost of POT for large-scale applications could be a direction for future optimization.
>
> **A1:** Thank you for your insightful suggestion. In Q4, we have validated PROTOCOL's computational cost on varying data scales (from 5K to 40K data points). The results demonstrate **near-linear growth**, showing PROTOCOL's potential for large-scale applications. In future work, we will continue to investigate and analyze POT's computational efficiency on datasets of even larger scale.
>
> **Q2:** Strengthen the logical connections between components.
>
> **A2:** Good suggestion!  We will enhance the logical structure of **Section 4 (Methodology)** as follows:
>
> **4.1 Motivation**. First, we will clearly articulate the two major challenges in imbalanced multi-view clustering, helping readers better understand the correspondence between challenges and their respective solutions.
>
> **4.2 Multi-view POT Label Allocation** (originally 4.1).  We will explicitly state at the beginning: "We propose a multi-view POT label allocation method that learns imbalanced class distribution of multi-view data through multi-view representation learning and a POT-based self-labeling mechanism."  Additional logical connections will be added between subsubsections to strengthen coherence. At the end, we will add a transition paragraph: "Through the learning of these components, PROTOCOL can effectively perceive the imbalanced distribution of multi-view data. This leads to the next challenge: how to mitigate representation degradation of minority samples, which we will address in next subsection."
>
> **4.3 Multi-view Class-rebalanced Contrastive Learning** (originally 4.2). We will first analyze the fundamental causes of representation degradation in minority samples, then introduce our solution.
>
> At the end of  **Methodology**, we will summarize how PROTOCOL systematically addresses the two challenges. Specifically, we will add the following description:"Imbalanced multi-view data is a more realistic application setting. PROTOCOL addresses the two key challenges of imbalanced multi-view data through POT self-label allocation and class-rebalanced contrastive learning."
>
> These modifications will make the logical connections between components more prominent.
>
> **Q3:** Convergence theory analysis.
>
> **A3:** Thank you for your constructive suggestion. Due to space limitations, we provide a brief theoretical analysis of convergence here. In the revised version, we will provide theoretical foundations for the convergence analysis.
>
> Our POT scaling algorithm extends the Sinkhorn-Knopp iteration by incorporating partial optimal transport with weighted KL divergence constraints. The algorithm achieves optimal label assignment through an efficient dual-form scaling iteration process. Based on [1], we prove that when $\epsilon, \beta > 0$, the algorithm guarantees linear convergence to a unique solution. The convergence rate depends on the entropic regularization parameter, weighted KL divergence weight, and cost matrix condition number. Our method introduces a dynamic mass parameter $\lambda$ for smooth transition from high-confidence samples to global optimal solutions. Moreover, experimental results validate both the efficiency and effectiveness of PROTOCOL, demonstrating its stability in handling imbalanced multi-view clustering.
>
> [1] Scaling Algorithms for Unbalanced Optimal Transport Problems (Mathematics of Computation 2018)
>
> **Q4:**  About PROTOCOL's computational cost scaling with dataset size.
>
> **A4:** Thank you for your valuable suggestion. Per your suggestion, we evaluated PROTOCOL's computational cost across four different data scales (5K to 40K samples) on the CIFAR10 dataset. As shown in **Figure A2** of the PDF file provided in the anonymous link (https://zenodo.org/records/15119555), the results demonstrate that PROTOCOL's computational cost scales **nearly linearly** with the number of samples. All experiments were conducted on an NVIDIA GeForce RTX 3090 GPU.
>
> **Q5:** About recent works on imbalanced multi-view clustering and suggested Ref [1].
>
> **A5:** Thank you for your suggestion. To the best of our knowledge, we are the first to systematically study the imbalanced multi-view clustering problem. While [1] is a single-view method for class imbalance problems, different from our multi-view approach, we will discuss it in our revised version.
>
> [1] Adaptive K-means clustering based under-sampling methods to solve the class imbalance problem (Data and Information Management 2024)

---

### Official Review · Reviewer_LTVa · 2025-03-14

**Overall Recommendation:** 3

**Summary:**

This paper proposes the first systematic study on the common class imbalance problem in multi-view clustering and develops a new framework called PROTOCOL. This method reformulates the imbalanced clustering problem as a partial optimal transfer problem by mapping multi-view features to a consensus space, and introduces step-by-step quality constraints and weighted KL divergence to perceive class imbalance. At the same time, class rebalanced contrastive learning enhanced by partial optimal transfer is used at the feature and category levels, combined with logit adjustment and category-sensitive learning, to alleviate the representation degradation problem of minority samples.

**Claims And Evidence:**

Yes

**Essential References Not Discussed:**

The paper comprehensively reviews the most relevant literature in the fields of Multi-View Clustering.

**Experimental Designs Or Analyses:**

Yes. The experimental setting and results have been reviewed.

**Methods And Evaluation Criteria:**

Yes

**Other Comments Or Suggestions:**

NA

**Other Strengths And Weaknesses:**

Strengths:
1. The paper innovatively integrates partial optimal transport with contrastive learning, utilizing progressive quality constraints and a weighted KL divergence to effectively perceive and model imbalanced distributions, while simultaneously enhancing the representation of minority samples at multiple levels.
2. Extensive experiments conducted on five datasets convincingly demonstrate the method’s superior performance in handling imbalanced multi-view data, providing robust empirical support for the proposed approach.

Weaknesses:
1. Although the paper targets imbalanced clustering, it does not clearly describe the specific operations involved nor adequately articulate the inherent challenges of imbalanced clustering in the motivation section.
2. The experiments are limited to datasets with a maximum scale of only 50,000 samples; the authors should consider validating their approach on larger-scale datasets.
3. In regard to Equation 28, which introduces the common semantic loss, the paper should provide a clearer explanation of its advantages and its specific impact on imbalanced clustering scenarios.

**Questions For Authors:**

NA

**Relation To Broader Scientific Literature:**

The key contributions of the paper are related to the broader scientific literature on Multi-View Clustering.

**Theoretical Claims:**

Yes. The methodology of the paper has been reviewed.

---

> ### Author Rebuttal · Authors · 2025-03-31
>
> We sincerely appreciate your recognition of our work's novelty as the first to identify and systematically study the class imbalance problem in multi-view clustering, as well as your positive feedback on our method's effectiveness and robustness. Furthermore, we are deeply grateful for your thorough review and constructive suggestions on our manuscript.
>
> **Q1:** The inherent challenges of imbalanced clustering.
>
> **A1:** Thank you for your constructive suggestion.  We will further clarify the two main challenges of imbalanced multi-view clustering (see lines 43 and 61) in the **Motivation** subsection of the revised version.
>
> We will add a **Motivation** subsection in the **Methodology** section to clarify the two key challenges and explicitly indicate how each module of our method addresses these challenges:
>
> **(1) How to perceive class imbalance distribution.** The challenge lies in detecting imbalanced distributions without labeled data in unsupervised settings. Existing methods, assuming uniform class distributions, often fail to handle imbalanced data effectively. This challenge will be addressed in **Multi-view POT Label Allocation** subsection.
>
> **(2) How to mitigate representation degradation of minority samples.** Minority samples, due to their scarcity, often receive insufficient attention during learning, resulting in poor feature representations that inadequately characterize their classes. This challenge will be addressed in **Multi-view Class-rebalanced Contrastive Learning**  subsection.
>
> This creates a more coherent flow from challenges to solutions, helping readers better understand both our motivation and technical approach.
>
> **Q2:** Validate our method on larger-scale datasets.
>
> **A2:** Thank you for your insightful suggestion. Per your suggestion, we validated our method on larger-scale dataset (CIFAR100 with 60,000 samples). As shown in **Figure A1** of the PDF file provided in the anonymous link  (https://zenodo.org/records/15119555), PROTOCOL achieves the best performance compared to other methods, demonstrating its effectiveness in handling class imbalance ($R$=0.1) on large-scale datasets.
>
> CIFAR100 is considered a large-scale dataset among those commonly used in multi-view clustering [1-4]. In future work, we will continue to explore PROTOCOL's potential on even larger-scale datasets.
>
> [1] A Comprehensive Survey on Multi-View Clustering (TKDE 2023)
>
> [2] Representation Learning in Multi‑view Clustering: A Literature Review (Data Sci. Eng 2022)
>
> [3] Differentiable Hierarchical Optimal Transport for Robust Multi-View Learning (TPAMI 2023)
>
> [4] Adversarially Robust Deep Multi-View Clustering: ANovel Attack and Defense Framework (ICML 2024)
>
>
>
> **Q3:** The advantages of Eq. (28) in imbalanced multi-view clustering scenarios.
>
> **A3:** Good suggestion! Eq. (28) employs contrastive learning to maintain semantic consistency across views for the same class. The denominator term $\sum_{j=1,j\neq i}^{K} \mathcal{D}(\{P_i^v, P_j^v\})$ measures negative pair similarities, ensuring comprehensive discrimination between classes. This design helps distinguish minority from majority classes while preserving cross-view semantic consistency, thereby enhancing representation learning for minority classes.
>
> In imbalanced multi-view clustering, cross-view semantic alignment is crucial due to minority classes' lower error tolerance. Unlike balanced scenarios where abundant samples can help correct semantic bias, minority classes have limited samples to rely on. Cross-view semantic alignment enables different views to complement each other, effectively reducing representation bias for minority classes.
>
> We will add a clear explanation of Eq. (28) in the revised version to highlight its advantages in imbalanced multi-view clustering scenarios.

---

> > ### Comment · Reviewer_LTVa · 2025-04-07
> >
> > Thanks to the author for the response, I decided to keep my rating unchanged.

---

> > > ### Author Response · Authors · 2025-04-08
> > >
> > > We appreciate your recognition of our work and thank you for your time and effort.

---

### Decision · Program_Chairs · 2025-05-01

**Decision:**

Accept (poster)

**Comment:**

The paper proposes PROTOCOL, a new framework for imbalanced multi-view clustering. All reviewers acknowledge the novelty of this work and the thoroughness of its experimental analysis, unanimously recommending a Weak Accept (WA) rating. Based on this consensus, I recommend accepting the paper.